# Unhealthy yet Avoidable—How Cognitive Bias Modification Alters Behavioral and Brain Responses to Food Cues in Individuals with Obesity

**DOI:** 10.3390/nu11040874

**Published:** 2019-04-18

**Authors:** Nora Mehl, Filip Morys, Arno Villringer, Annette Horstmann

**Affiliations:** 1Department of Neurology, Max Planck Institute for Human Cognitive and Brain Sciences, 04103 Leipzig, Germany; mehl@cbs.mpg.de (N.M.); morys@cbs.mpg.de (F.M.); villringer@cbs.mpg.de (A.V.); 2MaxNetAging Research School, 18057 Rostock, Germany; 3Leipzig University Medical Centre, IFB Adiposity Diseases, 04103 Leipzig, Germany; 4Leipzig University Medical Centre, Collaborative Research Centre 1052-A5, 04103 Leipzig, Germany; 5Department of Psychology and Logopedics, Faculty of Medicine, University of Helsinki, 00014 Helsinki, Finland

**Keywords:** cognitive bias modification, obesity, approach–avoidance task, fMRI

## Abstract

Obesity is associated with automatically approaching problematic stimuli, such as unhealthy food. Cognitive bias modification (CBM) could beneficially impact problematic approach behavior. However, it is unclear which mechanisms are targeted by CBM in obesity. Candidate mechanisms include: (1) altering reward value of food stimuli; and (2) strengthening inhibitory abilities. Thirty-three obese adults completed either CBM or sham training during functional magnetic resonance imaging (fMRI) scanning. CBM consisted of implicit training to approach healthy and avoid unhealthy foods. At baseline, approach tendencies towards food were present in all participants. Avoiding vs. approaching food was associated with higher activity in the right angular gyrus (rAG). CBM resulted in a diminished approach bias towards unhealthy food, decreased activation in the rAG, and increased activation in the anterior cingulate cortex. Relatedly, functional connectivity between the rAG and right superior frontal gyrus increased. Analysis of brain connectivity during rest revealed training-related connectivity changes of the inferior frontal gyrus and bilateral middle frontal gyri. Taken together, CBM strengthens avoidance tendencies when faced with unhealthy foods and alters activity in brain regions underpinning behavioral inhibition.

## 1. Introduction

The way we process and react to food cues might play an important role in the development and maintenance of unhealthy eating and obesity. It has been observed, for example, that overweight and obese individuals show an attention bias towards food images compared to healthy-weight controls and that obese participants display a food approach bias in comparison to lean participants [1,2,3,4]. This automatic and biased processing of often unhealthy food cues may contribute to the overconsumption of these foods [5,6], especially in current “obesogenic” environment.

Dual-process models address these automatic behavioral tendencies. The reflective–impulsive model [7], for example, states that during automatic behavior, the fast impulsive system overrules the slower reflective system. The former is hereby guided by previously formed associations—approach positive and avoid negative stimuli—while the latter relies on explicit knowledge [7,8]. The incentive sensitization theory (e.g., [9]) further states that through repeated exposure, a reinforcer (e.g., tasty food) acquires incentive salience qualities via the brain’s reward system. In consequence, associated stimuli become attention-grabbing and the motivation to approach them is increased [10].

These theories might explain a paradox in maladaptive eating and addictive behaviors, where individuals continue to behave in disadvantageous ways despite better knowledge and often against personal intentions. Heavy drinkers, for example, were found to crave alcohol without necessarily liking it [11,12] and were repeatedly found to display an approach bias towards alcohol [8,13,14].

Cognitive bias modification (CBM) presents a tool that aims at changing these automatic tendencies towards problematic stimuli, subsequently improving maladaptive behavior. Generally, CBM refers to a class of interventions that use experimental paradigms to change biased cognitive processing. This can be done in various domains such as attention, association, and behavioral approaches or avoidance tendencies [15]. Promising effects have been shown, for example, in alcohol-dependent subjects and smokers [8,16,17], where a CBM intervention decreased approach bias towards problematic stimuli while reducing consumption thereof. Regarding eating behavior, research has produced mixed findings [18]. Approach bias for and consumption of chocolate could be decreased through CBM in normal-weight individuals [19]. In contrast, no CBM effects were found in normal-weight females across three studies for both unhealthy food and chocolate [20]. In a sample of lean and obese participants, CBM reduced approach bias towards unhealthy food only in obese individuals [4]. Discrepancies in these results, however, might be explained by differences in samples and stimuli. Approach behavior towards chocolate or unhealthy food, for example, might not affect normal-weight females the same way as individuals with obesity. In turn, they might be differently affected by a CBM intervention aimed at changing food approach behavior.

Despite the indicated evidence for CBM’s efficacy in the behavioral context, not much research has been conducted regarding underlying neural mechanisms. One study investigated neural correlates of CBM in hazardous drinkers, where half of the participants received CBM while the other half received sham training. CBM was associated with reducing activity in the medial prefrontal cortex [21]. This structure, together with the nucleus accumbens and the posterior cingulate gyrus, is engaged in approach behavior towards problematic stimuli [22,23].

In this study we investigated neural correlates of CBM in obesity by applying an approach-avoidance training [8] in obese individuals in the fMRI scanner, randomly assigning participants to a training or a sham-training condition. On the behavioral level, the training was hypothesized to induce two effects: decreased approach tendencies towards unhealthy foods, and increased approach tendencies towards healthy foods. On the neural level, we hypothesized that CBM training could work through two mechanisms, by: (1) changing rewarding values of food stimuli and activation in reward-related regions; and (2) increasing inhibitory abilities and changing activity of brain regions engaged in inhibitory processing and cognitive control. We further explored whether CBM training induces differences in task-independent resting-state functional connectivity.

## 2. Materials and Methods

### 2.1. Participants

Thirty-three obese participants (18–35 years) took part in the experiment (mean body mass index (BMI) = 36.49 kg/m^2^, σ = 6.29, mean_age_ = 29.5 years, σ = 4.5; for sample characteristics see Table 1). Sample size was selected according to a similar study investigating CBM effects in alcohol-dependent patients [23]. Participants met the following inclusion criteria: right-handed, no history of neurological/psychological diseases, no thyroid disease, normal or regulated to normal blood pressure, no drug/alcohol addiction, no smoking, and no MRI-related contraindications. Volunteers received monetary compensation. Participants were randomly divided into a training (*n* = 17) and a sham-training group (*n* = 16) and were not informed and not aware that any training would take place. The experiment introduction was performed by a blinded experimenter. The study was conducted according to the Declaration of Helsinki and approved by the Ethics Committee at the University of Leipzig. All participants gave written informed consent prior to the study.

### 2.2. Behavioral Assessment

BMI was calculated before the experiment, after height and weight of the participants were measured by the experimenter. Before and after the MRI part, participants rated their mood, hunger, and tiredness on visual analogue scales (VAS, 0–10, see Table 1). Subjects were further asked to sort forty credit card-sized food pictures on a cardboard with a scale (0–10). Subjects were instructed to sort pictures in line with how healthy or unhealthy they perceive them and how much they like or dislike the depicted food items. This measure was included to assess explicit evaluations of food stimuli regarding healthiness and liking. In order to assess whether CBM would affect these ratings, the picture-sorting task was done before and after the experimental paradigm. The picture set was independent of the one used in the fMRI task. It included healthy and unhealthy food pictures of comparable healthiness and liking to the fMRI picture set (see Section 2.5: Selection of stimuli for details).

### 2.3. Questionnaire Measures

We assessed eating behavior on the three dimensions: cognitive restraint, disinhibition, and hunger with the Three Factor Eating Questionnaire (TFEQ [24]). To evaluate to what extent a person’s behavior is driven by reward and punishment we used the behavioral inhibition/activation system scales (BIS/BAS [25]). Questionnaires were used as baseline comparison of groups included in the study. Distribution of all questionnaire measures was normal or close to normal and equal variance between groups was assumed for all the tests (for details see Appendix A).

### 2.4. fMRI Task

We used a training version of the approach–avoidance task (AAT; described in [4], for task details see Figure 1A), which measures and modifies [4] automatic approach and avoidance tendencies towards unhealthy and healthy food pictures (30 healthy/30 unhealthy). Participants react with push and pull movements of a joystick to the format of presented food pictures (push–vertical/pull–horizontal, or push–horizontal/push–vertical). The association between the picture format and joystick movement was randomly assigned to each participant and did not change throughout the task. The AAT consisted of three main phases: a pre-phase, a training or a sham-training phase, and a post-phase. In the pre-, post-, and sham training phases, food pictures were presented equally often in push and pull formats. In the training phase 90% of unhealthy pictures appeared in a push format, and 90% of healthy pictures appeared in a pull format (Figure 1B). In this respect there was no modification of the task as compared to [4]. The amount of approach and avoidance responses was equal in both groups—any behavioral effects would therefore be related to the pictures’ content.

Further, we tested whether potential changes in automatic action tendencies are specific to pictures included in the training phase or generalized to the entire category (healthy vs. unhealthy). Therefore, only a subset of pictures used in pre- and post-phases was used for training (randomly chosen set of 20 out of 30 pictures for each participant).

The AAT lasted around 40 min and was symmetrically divided into four runs, each including 110 trials (independent of pre-, training, or post-phases). Participants were offered breaks between runs to relax and close their eyes.

### 2.5. Selection of Stimuli

Food images for the AAT and the picture-sorting task were selected from the food-pics database [26] and categorized into healthy and unhealthy according to [4]. The categorization was done in line with nutritional values and guidelines, e.g., nutritional tables [27], the Healthy Eating Index [28], or the Dietary Guidelines for American Adherence Index [29]. Healthy food items hereby met at least three of the following five criteria: (1) high nutrient density, (2) low energy density OR high amount of monounsaturated fatty acids, omega-3 fatty acids, or omega-6 fatty acids, (3) high fiber content and/or high water content, (4) low amount of added sugar, and (5) low salt content. Unhealthy food items met at least one of the following criteria: (1) low nutrient density combined with high energy density, (2) high energy density AND an unfavorable ratio of saturated fatty acids, omega-3 fatty acids, or omega-6 fatty acids, and (3) high amount of added sugar. Further, a pre-rating online survey was conducted in order to validate nutritional categorization from a subjective perspective. Participants (*n* = 98, BMI range from 18 to 36 kg/m^2^) were asked to rate food pictures on a scale from 0 to 10 (0 = unhealthy, 10 = healthy) regarding how healthy or unhealthy they perceived them to be. Only images that were clearly identified as healthy or unhealthy were included (mean ratings between 0 and 2 or 8–10). This way, pictures were categorized as being healthy or unhealthy according to both objective nutritional and subjective criteria.

### 2.6. Neuroimaging

We collected resting-state (pre-AAT and post-AAT), task-related and anatomical neuroimaging data. Data were acquired using a 3T Siemens SKYRA scanner with a 20-channel head coil. For the AAT, 1104 T2*-weighted images were collected (echo time TE = 22 ms, flip angle FA = 90°, repetition time TR = 2000 ms, 40 slices, voxel size: 3.0 × 3.0 × 2.5 mm^3^, distance factor: 20%, field of view FoV: 192 × 192 mm^2^, ascending order). The 2*320 open-eyes resting-state T2*-weighted images were acquired using the same parameters. A high-resolution anatomical magnetization prepared rapid gradient-echo (MPRAGE) image was acquired for each participant (TE = 2.01 ms, FA = 9°, TR = 2300 ms, inversion time TI = 900 ms, voxel size: 1 × 1 × 1 mm^3^, distance factor: 50%, FoV: 256 × 256 mm^2^).

### 2.7. Data Analyses

#### 2.7.1. Behavioral Analysis

Mean reaction times were calculated for both picture categories (healthy/unhealthy food) and for both conditions (avoid/approach) during all three phases of the experiment. Bias scores were calculated as difference scores per category and condition: (healthy_push–healthy_pull) and (unhealthy_push–unhealthy_pull). Positive scores reflect faster approach reactions for the respective food category, while negative scores indicate faster avoidance reactions.

No subject had to be excluded due to outliers or error rate. Outliers were defined as mean reaction times below or above 2 standard deviations from the group mean. The task was performed with high accuracy (mean = 97%, SD = 3.23%).

During the pre-phase, bias scores significantly differing from zero would reflect baseline behavioral tendencies. Further, to ensure that no baseline group differences were present, we compared bias scores of training and the sham-training group. Analyses were carried out using one-sample and independent-samples *t*-tests, respectively.

Changes from pre to post were analyzed using a 2 × 2 × 2 repeated measures ANOVA (rmANOVA). Group (training/sham-training) was used as a between-subject factor, and image category (healthy/unhealthy) and time (pre/post) as within-subject factors. We followed up by testing if bias scores significantly differed from zero in the post-phase with *t*-tests.

#### 2.7.2. fMRI Data Analysis

##### Data Preprocessing

AAT-fMRI and rsfMRI data were preprocessed in a similar fashion. Data were preprocessed and statistically analyzed using FMRIB Software Library 5.0.8 (FSL, The University of Oxford, Oxford, United Kingdom [30]), SPM 12 revision 6225 (Wellcome Department of Cognitive Neurology, London, United Kingdom), Analysis of Functional NeuroImages version 17.0.04n (AFNI, [31], National Institute of Mental Health Scientific and Statistical Computing Core, Bethesda, MD, US), Advanced Normalization Tools (ANTs [32]), and MATLAB R2012b (The MathWorks, Inc., Natick, Massachusetts, United States). Firstly, to enable further preprocessing steps, high-resolution anatomical images were skull-stripped using the FSL’s brain extraction tool [33] and the SPM 12 segmentation tool. Functional data were motion-corrected using McFLIRT [34], fieldmap-corrected and registered to high-resolution anatomical images (FLIRT, boundary-based registration [34,35,36]), slice-timing corrected, and smoothed with a 6-mm full width at half maximum (FWHM) Gaussian kernel (not rsfMRI data used for connectivity analysis [37]) using the FSL’s FMRI Expert Analysis Tool (FEAT). To ensure that motion- and physiological noise-related artefacts were removed from the functional time-series, we used the independent component analysis automatic removal of motion artefacts (ICA AROMA, [38]) toolbox on the time-series. Further, we regressed out the signal in white matter and cerebrospinal fluid from the functional data. Then, anatomical images were normalized to a 3-mm Montreal Neurological Institute (MNI) template using ANTs. Using transformation information from the previous registration steps, functional images were registered to the 3-mm MNI template using ANTs. Prior to statistical analysis on an individual level, AAT fMRI data were high-pass filtered with a filter of 128 s (SPM). Prior to connectivity analyses, resting state data were high pass filtered (FSL; σ = 22).

##### AAT fMRI Data Analysis

A random-effects analysis was performed using SPM12. Regressors on a subject level were entered into a general linear model (GLM) and convolved with a double-gamma hemodynamic function. Contrast files were entered into second-level analysis, where we compared subjects as groups. BMI and age were entered into the analysis as covariates of no interest. By entering BMI as a covariate, we investigated general group effects dependent on training condition only and accounted for between-subject differences that could potentially be caused by differences in BMI. Age was entered as a covariate since groups differed in this respect. Results were thresholded at a whole-brain voxel-wise level with a threshold of 0.005 and on a cluster level with an FWE-corrected threshold of 0.05.

##### GLM1: Pre and Post Data Analysis

On a single-subject level we entered pre and post trials into a general linear model. This resulted in 16 different regressors over two sessions. These included 4 regressors for the picture presentation period pre-training (healthy_pull, healthy_push, unhealthy_pull, unhealthy_push), 4 regressors for the zooming period pre training (corresponding to four different types of trials), and a similar set of 8 regressors for the post-phase. For the picture presentation period, onsets of the regressors were time-locked to the picture presentation, and event duration was equal to the reaction time. This variable epoch model was described as the most appropriate for reaction time tasks [39]. The onsets of the zooming period regressors were time-locked to the end of the picture presentation period, and the durations were set to 750 ms.

We investigated neural correlates of food approach and avoidance tendencies, as well as training effects on the brain. For the pre-phase, first-level contrasts included general food approach and avoidance, and the same contrasts specific to each food category. We further compared food avoidance separately for healthy and unhealthy food pictures within and between groups. To elucidate neural correlates of food approach tendencies independent of group membership, individual contrasts during the pre-phase of all participants were entered into a one-sample *t*-test on the second level. To ensure that no pre-training differences in task-related brain activity between groups were present, we entered individual contrasts into a two-sample *t*-test. Further, contrasts involving comparisons of pre- and post-phases were entered into two sample *t*-tests to investigate effects of training.

##### GLM2: Psychophysiological Interactions Analysis

To investigate whether our intervention was related to changes in functional connectivity, we conducted a psychophysiological interactions (PPI) analysis. This analysis compares brain connectivity changes from a specified seed in the brain between two different experimental conditions. Firstly, we defined a 6-mm sphere (radius) around the peak voxel in a cluster reflecting training effects (right angular gyrus). Secondly, we extracted raw time-series from this volume of interest (VOI). Thirdly, we defined a new GLM consisting of six different regressors: the time course of the VOI pre training (physiological factor), the main effect of unhealthy_push condition (psychological factor), the interaction term between the two factors for the pre training phase, and three corresponding regressors for the post training phase.

##### Resting-State fMRI Data Analysis

We acquired and analyzed resting state data in order to investigate whether the effects of CBM are transferrable to functional changes in the brain not directly related to performance during the AAT. These data can be used to analyze resting state functional connectivity, answering the question of how different brain regions interact with each other. Resting state connectivity analysis helps to understand how all brain regions generally interact with each other (degree centrality, DC), but also how specific a priori defined brain regions correlate with other brain areas (seed-based connectivity analysis, SCA). To our knowledge, this is the first study investigating this particular aspect. Hence, we decided to use a hypotheses-based approach of seed-based connectivity analysis with different inhibitory and reward-related seed regions, and also a hypotheses-free approach of degree centrality.

##### Seed-Based Connectivity Analysis

We investigated seed-based connectivity using predefined regions of interest as seeds (ROI, see section “seed definition”). An unsmoothed [37] functional time series was entered into the analysis. Analysis was performed using Nipype and Nilearn algorithms. It resulted in 16 connectivity maps—one for each ROI, and one for each phase of the experiment (pre- and post-AAT)—which were then smoothed with a 6-mm FWHM Gaussian kernel. Pre-AAT maps were subtracted from the post-AAT maps, and resulting maps were entered into a two-sample *t*-test to investigate group differences using FWE-corrected and Bonferroni-corrected (number of seeds) statistical thresholds.

##### Seed Definition

To investigate task-unrelated connectivity differences caused by CBM, we defined a number of seeds directly related to reward processing, visual food stimuli processing, and inhibitory control. This was done in order to test our hypotheses of reward vs. inhibitory mechanisms involved in the CBM. The following seeds were included in our study: the medial and the left and right dorsolateral prefrontal cortex (mPFC, dlPFC, respectively; coordinates from [40]), the left and right amygdala and nucleus accumbens (Amy, NAcc, respectively; coordinates from pickatlas [41]), and the left middle frontal gyrus (MFG; coordinates from [42]). The mPFC, Amy, and NAcc were previously shown to be engaged in approach-avoidance tendencies and are widely accepted reward-related brain regions [23]. The mPFC is widely accepted as the brain’s valuation center [43]. The amygdala is important for Pavlovian learning and formation of emotional memories [44,45], whereas the NAcc receives and sends dopaminergic projections in response to rewarding stimuli [44,45,46,47]. The dlPFC was previously related to approach-avoidance tendencies and is an inhibitory brain region [48,49]. Lastly, the left MFG is a region preferentially activated for viewing high versus low caloric food stimuli [42].

##### Degree Centrality

Degree centrality denotes a number of direct connections of a node to all other nodes in the network [50]. We used AFNI within the Nipype framework, with correlation thresholds of 0.5 (only *r* >0.5 was included in results). Firstly, we calculated DC maps separately for each participant and phase of the experiment. Secondly, similarly to SCA analysis, we subtracted the pre-AAT maps from the *post* AAT maps and entered resulting volumes into a two-sample *t*-test.

## 3. Results

### 3.1. Behavioral Results

There were no baseline group differences regarding hunger, tiredness, mood, or questionnaire measures (smallest *p* = 0.082, Table 1 and Appendix A), and subjective ratings of the AAT stimulus material (smallest *p* = 0.498, Appendix A). All participants rated healthy images as healthier (t(32) = 36.723; *p* <0.001, d = 6.40) and liked them more (t(32) = 5.507; *p* <0.001, d = 0.96) than unhealthy images.

To test if CBM would affect explicit ratings of the independent picture set of the sorting task (Appendix A), we performed separate 2 × 2 × 2 rmANOVAs for healthiness and liking ratings with image category, time and group as factors. They revealed no interactions with group (healthiness: F(1,31) = 0.001, *p* = 0.975, η^2^_p_ = 0.000; liking: F(1,31) = 0.453, *p* = 0.503, η^2^_p_ = 0.014), indicating no effects of CBM on this task. This was confirmed by a Bayesian rmANOVA (JASP version 0.9; JASP Team 2018) with identical factors—a lack of differential group effect for healthiness and liking ratings was 2.886 and 3.847 times more likely than the alternative explanation, respectively.

#### AAT

We first tested for baseline food approach bias. As expected, both groups showed a significant food approach bias, as bias scores for the healthy and unhealthy images differed from zero (training group: t(16) = 2.994, *p* = 0.009 and t(16) = 2.334, *p* = 0.033; sham-training group: t(15) = 3.728, *p* = 0.002 and t(15) = 2.218, *p* = 0.042, respectively). There were no group differences in approach bias (healthy: t(31) = 0.557, *p* = 0.581; unhealthy: t(31) = −0.465, *p* = 0.645).

Our main question was whether CBM affects approach behavior towards food stimuli and whether this depends on picture category. We used a 2 × 2 × 2 rmANOVA with group, image category and time as factors. We found a significant three-way interaction of group (training or sham-training), image category (healthy vs. unhealthy) and time (pre- vs. post-training; F(1,31) = 8.902, *p* = 0.006, η^2^_p_ = 0.223). Follow-up paired *t*-tests indicated that the training group, as opposed to the sham-training group, had decreased approach tendencies towards unhealthy images (Table 2, Figure 2), driven by significantly faster push movements for unhealthy images after the training (t(16) = 2.735, p = 0.015).

To test for generalization, a 2 × 2 × 2 rmANOVA was performed in the training group. Factors included picture set (trained vs. not-trained images), image category, and time. A significant three-way interaction would indicate lack of generalization, showing that bias scores for trained and not-trained images of the same category were not similarly affected by the training. The three-way interaction was marginally not significant (F(1,15) = 4.464, *p* = 0.051, η^2^_p_ = 0.218), suggesting that generalization might have occurred. A follow-up Bayesian rmANOVA with identical factors showed that a model with the three-way interaction, compared to a model without this interaction was 0.493 less likely, indicating anecdotal evidence in favor of generalization. We therefore cannot conclude with certainty whether generalization occurred.

### 3.2. Neuroimaging Results

#### 3.2.1. GLM1

##### Baseline Food Approach and Avoidance

We investigated neural correlates of pre-training approach and avoidance tendencies with a one-sample t-test, as in this stage groups did not differ in any way. Contrasts included food approach and food avoidance, together and separately for healthy and unhealthy food. A contrast corresponding to general food avoidance (push > pull independent of picture category) revealed significant clusters in the right angular gyrus (rAG) and the cuneus (Figure 3A). Food avoidance activations were driven by the unhealthy food category (push > pull for unhealthy food). For the general approach for food (pull > push), we found a significant cluster in the left postcentral gyrus (Figure 3B, Table 3). We found no further significant results and did not find group differences for above-mentioned contrasts.

##### Pre- to Post- Changes

As the main analysis of interest, we tested whether the training effect—decreased approach bias towards unhealthy foods—was associated with neuronal changes. We contrasted unhealthy food avoidance (unhealthy_push > unhealthy_pull) with healthy food avoidance (healthy_push > healthy_pull) before vs. after training. In the training group, the rAG showed decreased activity post-training, whereas the left middle occipital gyrus showed increased activity (Figure 3). We then compared unhealthy food conditions (unhealthy_push > unhealthy_pull) pre- and post-training and found a similar effect. Here, we also found decreased activity in the left lingual gyrus for the sham-training group, and a group difference in the cuneus (training > sham-training group). The effect in the rAG was driven by lower brain activity in the training group for the post- vs. pre-phase for unhealthy food avoidance (unhealthy_push > unhealthy_pull). Results indicate that brain activity in the right rAG for pushing unhealthy foods in the training group decreased after the training (Table 3).

#### 3.2.2. GLM 2: PPI Analysis

We consistently found the rAG to be associated with unhealthy food avoidance and the effects of CBM, and therefore performed PPI analysis with the rAG as the seed. We compared connectivity differences for unhealthy food avoidance between pre- and post-phases. This analysis showed a significant cluster in the rSFG/rMFG and in the right caudate/putamen, indicating that connectivity between the rAG and these structures increased post-training in the training group (Table 3, Figure 3).

#### 3.2.3. Resting-State Data

##### Seed-Based Connectivity Analysis

We investigated whether training induced connectivity changes between previously specified seed regions using resting-state, task-independent data. For the left MFG, we found a significant group*time interaction in the right MFG. We observed a similar interaction effect for connectivity between the left NAcc and the left inferior frontal gyrus (IFG; Table 3, Figure 3).

##### Degree Centrality

Similar to SCA, DC describes task-independent connectivity changes within the brain. These changes, however, are general and not specific to chosen ROIs. In our study this analysis did not produce any significant results.

## 4. Discussion

We investigated the underlying neural mechanisms of CBM in obese individuals. To this end, a training form of the AAT was applied in the fMRI scanner, where half of the participants received training, while the other half underwent sham training. This between-group design combined with fMRI measures can provide insight into whether the effects of CBM are mediated by: (1) changing rewarding values of food stimuli and brain activation in reward-related brain regions; or (2) increasing inhibitory abilities and affected brain regions engaged in inhibitory processing and cognitive control.

We firstly found that all participants displayed faster approach than avoidance reactions towards healthy and unhealthy food images, suggesting that approaching food could be a rather automatic process. This was paralleled by our findings on the neural level, where the right angular gyrus showed higher activation when participants had to avoid food—a potentially conflicting situation. The rAG is a part of the temporoparietal junction (TPJ), which is often related to both processing of social cues and attentional processes [51,52]. We further observed that CBM specifically affected the training group, as only in this group approach tendencies towards unhealthy food were successfully decreased. This was paralleled by a now lower activation in the rAG after training. Additionally, we observed group-specific changes in resting-state connectivity between inhibitory regions such as the MFG or the IFG [53,54,55,56] and in task-related connectivity between the rAG and the right caudate/putamen (dorsal striatum). This further supports the notion of avoiding food being a potentially conflicting situation, thus requiring activation of inhibitory and conflict resolution brain mechanisms. One of the possible mechanisms by which CBM works is a decrease of this demand by means of strengthening connectivity between inhibitory brain regions, as our study might suggest. Further, we found little evidence for altered reward valuation of food stimuli after CBM in both behavioral and imaging data. Specifically, healthiness and liking ratings of the AAT-independent picture set were not altered by the training, showing lack of evidence for altered reward valuation. Additionally, only one reward-related brain region was found in our imaging analysis in the seed-based connectivity analysis, namely the nucleus accumbens.

As previously mentioned, avoiding food was paralleled by higher activity of the rAG—a part of the TPJ. Bzdok and colleagues showed that the right TPJ links two brain networks integrating external (sensory) vs. internal (memory, social-oriented stimuli) information [51]. It is conceivable that a conflict between external instructions (avoid unhealthy food), and internal impulses (approach unhealthy food) increases activity of the rAG in order to solve this conflict. This is consistent with studies showing that the rAG is directly engaged in resolution of stimulus–response conflicts, but also attentional reorientation and response inhibition [57,58,59,60,61,62].

Though approach behavior towards unhealthy food pictures decreased in the training group, approach behavior towards healthy food pictures remained unchanged. This is in line with previous findings, where decreasing approach behavior towards unhealthy food appeared to be a main effect of CBM [4,63]. In our study, decreasing approach tendencies towards unhealthy food were paralleled by decreased brain activation in the rAG in the training group, although no group interaction was present for this effect. This lack of interaction, however, might be related to a low sample size and low statistical power (see bar plot in Figure 3C for more details). This suggests that training makes avoiding food a less conflicting and more automatic behavior. We cannot make inferences, however, on whether these training effects are specific to food images included in the training or if they could generalize to new and untrained food images. Our data provided no conclusive evidence for generalization effects. While generalization was repeatedly observed in other contexts, such as alcohol (e.g., [8]), results in the obesity context have generally been mixed [4,63]. One study, for example, found that training effects generalized to new, untrained food images [63], in that the reactions to these images changed in the training-induced direction. A different study with a similar design found no such effects [4]. Here, training-induced changes could only be observed in the reactions towards trained images.

For unhealthy food avoidance, we found increased post-training task-related connectivity between the rAG and the dorsal striatum, which is related to stimulus–response learning, executive attention and exerting cognitive control [64,65,66,67,68]. Increased connectivity between the dorsal striatum and rAG was previously related to explicit usage of learned stimulus–response-outcome associations [69]. This could be interpreted as a complementary effect of CBM training. Decreased activity of the rAG was paralleled by more efficient inhibition of automatic reactions after training, possibly through its increased coupling with other brain structures. This suggests that the rAG does not require similar activation strength as pre training. This interpretation is supported by resting-state connectivity results, where we found higher post-training resting-state connectivity between inhibitory regions in the brain—the bilateral MFG, and left IFG. The left MFG was previously shown to be activated for viewing high caloric food pictures [42]. Changed connectivity between the left and right MFG, structures engaged in response inhibition [53,54,55,56], might therefore be associated with training-induced stronger inhibitory tendencies towards unhealthy food pictures. We also found higher post-training connectivity of a reward-related region—the nucleus accumbens—and the left IFG, also engaged in response inhibition [70]. This could be associated with an increased inhibition of approach responses to rewarding stimuli. However, we also acknowledge the possibility that this relationship represents a bottom-up gating of relevant stimuli to the PFC.

In comparison to the alcohol context, where effects of CBM seem to be mediated by altering the rewarding properties of problematic stimuli, underlying neural mechanisms of CBM seem to differ in the food context. This might be explained by differences in the duration of training. Changes in stimuli evaluation and reward-related areas, as found in the alcohol context, are more of a reflective process, thus requiring longer training. However, a previous intervention study in obese individuals, applying food response inhibition training over a four-week period, found decreased brain activations in the insula, inferior parietal lobe, and putamen [71]. It seems that a longer training in the food context elicits changes in brain areas similar to those in our study. Hence, an alternative explanation of discrepancies between results in the food and alcohol context is that training in the food context works in a different way, possibly because food biases may depend on a different neural network.

It is important to consider limitations of this study. Firstly, sample size was moderate (33 participants). Secondly, CBM effects did not translate to the picture-sorting task, which aimed to assess explicit evaluations of food stimuli. This could point towards CBM training inhibition rather than affecting evaluation processes. This lack of transfer from implicit training to explicit evaluation is in correspondence with the lack of neuronal effects in valuation areas. Further, our training only focused on healthy and unhealthy categories without additional divisions, e.g., into sweet and savory. This may decrease the sensitivity of our analyses and may be related to lack of effects on the picture-sorting task. Also, we compared approach and avoidance tendencies between healthy and unhealthy food images, not including neutral images of non-food objects. Further, due to ethical reasons we did not train obese participants to approach unhealthy foods and avoid healthy food cues. Importantly, we do not show effects of training on food intake, which was not assessed in this study. This is a main goal of CBM studies and has only been investigated to a small degree [18,72]. Future studies should implement CBM interventions in real-life settings, assessing its impact on eating behavior.

However, our study provides a basis for future studies on food approach bias modification, which could focus on specifically strengthening inhibitory control, especially regarding unhealthy food. Additionally, showing that already one CBM session can modify approach tendencies in the laboratory context is very promising. In conclusion, we were able to show that obese individuals have automatic approach tendencies towards food. We further present a possibility to retrain and decrease approach tendencies, especially towards unhealthy foods, and give insight into underlying neural mechanisms. This study could constitute a basis for intervention programs utilizing similar behavioral paradigms. We suggest that these studies implement longer training periods, similar to ones used with alcohol-dependent patients. Further, training should specifically aim at strengthening inhibition, specifically towards unhealthy food, rather than encouraging approach towards healthy food, as often done in weight loss programs. Additionally, by showing neural correlates of CBM, our results contribute to possible brain stimulation research focusing on decreasing approach bias towards food.

## Figures and Tables

**Figure 1 nutrients-11-00874-f001:**
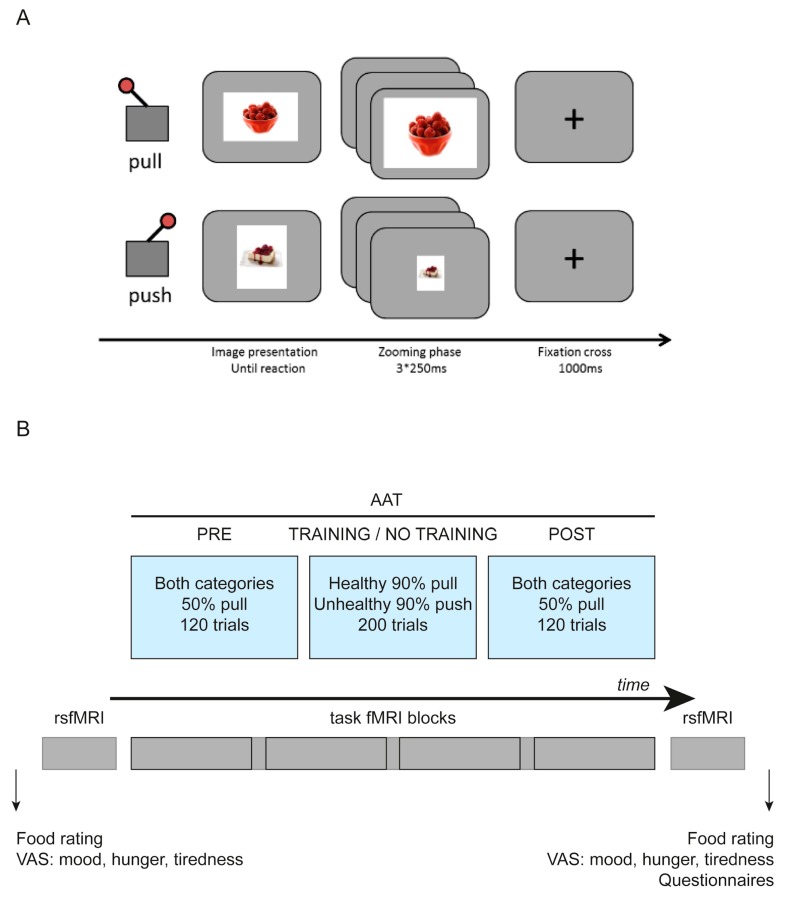
(**A**): Modified approach–avoidance task. (**B**): Overview of the experimental paradigm. AAT: approach–avoidance task, rsfMRI: resting-state functional magnetic resonance imaging.

**Figure 2 nutrients-11-00874-f002:**
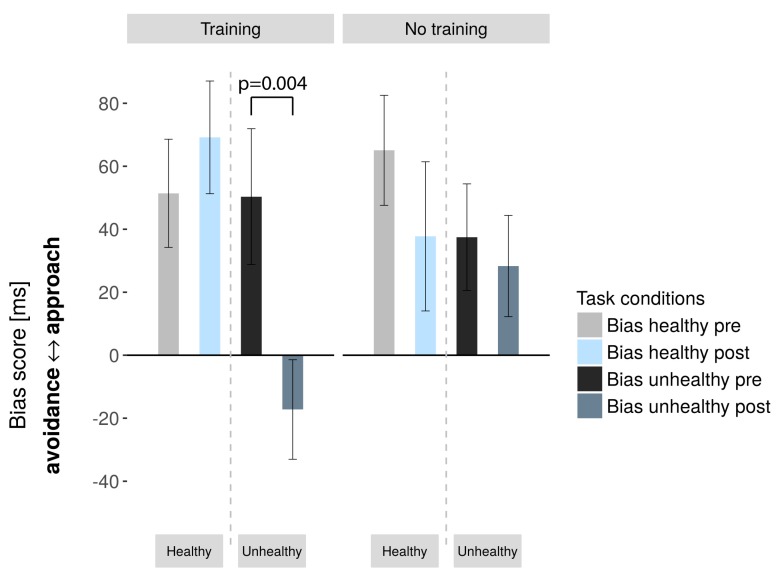
Bias scores in the training and sham-training groups pre- and post-training (error bars: standard error of the mean). We observed a significant three-way interaction between group, time, and image category.

**Figure 3 nutrients-11-00874-f003:**
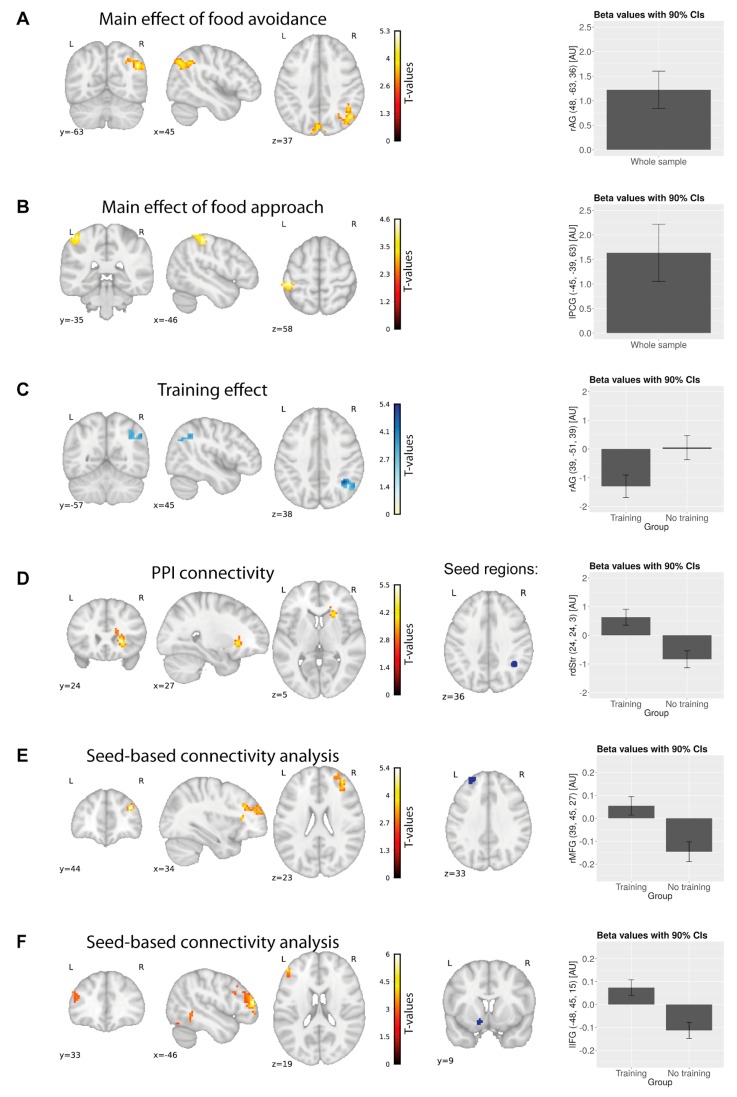
Main effects of food approach/avoidance pre-training in both groups together (**A**,**B**), and effects of cognitive bias modification (CBM) training along with contrast estimates (arbitrary units); note a different scale in the parameter estimates for subfigures. (**A**): Main effect of food avoidance. (**B**): Main effect of food approach. (**C**): Training effect was reflected in a decreased brain activity in the right angular gyrus for healthy food avoidance vs. unhealthy food avoidance. (**D**): Higher task-related connectivity between the right dorsal striatum and the right angular gyrus in the unhealthy push vs. unhealthy pull condition after training. (**E**): Increased connectivity post-training in the training vs. sham-training group in the left and right middle frontal gyri (resting-state seed-based connectivity analysis). (**F**): Increased connectivity post-training in the training vs. sham-training group in the left nucleus accumbens and inferior frontal gyrus (resting-state seed-based connectivity analysis). rAG: right angular gyrus; lPCG: left postcentral gyrus; rdStr: right dorsal striatum; rMFG: right middle frontal gyrus; lIFG: left inferior frontal gyrus; AU: arbitrary units; CI: confidence intervals.

**Table 1 nutrients-11-00874-t001:** Sample characteristics of the training and sham-training groups; *p*-values reflect significance of group differences. VAS: visual analogue scale; SD: standard deviation; BMI: body mass index.

	Training Group	No-Training Group	*p*-Value/T(31) Value (Unless Otherwise Indicated)	Effect Size |D| (Unless Otherwise Indicated)
	Mean/SD (unless otherwise indicated)			
*n*	17	16		
Sex	11 ♀. 6 ♂	7 ♀. 9 ♂	0.227/χ^2^ = 1.460	φ = 0.043
Age (years)	28/5	31/4	**0.027/2.314**	0.663
BMI (kg/m^2^)	35.57/4.63	36.95/7.63	0.530/0.635	0.219
Hunger (VAS cm; not hungry–hungry)	2.31/1.81	2.73/1.91	0.534/0.629	0.226
Tiredness (VAS cm; not tired–tired)	4.31/2.60	4.00/2.30	0.726/-0.354	0.126
Mood (VAS cm; in a bad mood–in a good mood)	7.69/1.58	8.20/1.26	0.711/ = 0.374	0.357

**Table 2 nutrients-11-00874-t002:** Bias scores for healthy and unhealthy images in the training and sham-training group for the pre- and post-phases. *p*-values reflect significance of changes from pre to post in bias scores.

Image Category	Training Group				Sham-Training Group			
	Pre	Post	*p*-Value/	Effect Size d	Pre	Post	*p*-Value/	Effect Size d
	Mean/SD		t(16)-Value		Mean/SD		t(15)-Value	
Unhealthy	50.35/	−17.24/	**0.004**/	0, 81	37.50/	28.31/	0.585/	0, 14
	88, 97	65, 18	−**3336**		67, 63	64, 34	0, 559	
Healthy	51.41/	69.18/	0.429/	0, 197	65.06/	37.75/	0.126/	0, 403
	70, 8	73, 78	−0, 812		69, 81	94, 76	1, 61	

SD: standard deviation. Bold values indicate statistically significant differences.

**Table 3 nutrients-11-00874-t003:** Brain regions showing training-related changes between- and within-groups.

Contrast (Pre > Post Phase)		Region of the Peak Voxel	Cluster Size (Voxels)	Coordinates (MNI)			Peak z Score	Peak *t* Score
Brain regions associated with baseline food approach/avoidance bias								
Food avoidance		Angular gyrus R	178	48	−63	36	4, 4	5, 28
		Cuneus	131	0	−87	24	4, 21	4, 96
Food approach		Postcentral gyrus L	98	−45	−39	63	3, 98	4, 62
Unhealthy food avoidance		Angular gyrus R	129	51	−66	33	4, 51	5, 45
		Cuneus	212	−3	−87	24	4, 1	4, 8
Brain regions showing training-related changes between- and within-groups								
Unhealthy food avoidance > healthy food avoidance	Training group	Angular gyrus R	99	51	−69	33	4, 06	4, 77
		Middle occipital gyrus L	163	−21	−90	−15	−4, 26	−5, 07
Unhealthy food avoidance	Training group	Inferior parietal lobe R	124	39	−51	39	4, 46	5, 41
	Sham-training group	Lingual gyrus L	202	−3	−75	9	3, 99	4, 65
	Training>sham-training	Cuneus L	163	−15	−75	9	−3, 73	−4, 27
Brain region showing increased activity for unhealthy food avoidance *pre-*training in the training group only								
Unhealthy food avoidance: pre-phase		Angular gyrus R	97	51	−66	30	4	4, 67
PPI connectivity differences in the training group from *pre-* to *post-*phase for unhealthy food avoidance								
PPI connectivity in the training group; seed: right angular gyrus		Putamen R	170	24	24	3	4, 51	5, 53
Regions showing a group by time interaction in the resting-state measures of brain activity and connectivity;								
SCA, left middle frontal gyrus		Middle frontal gyrus R	182	39	45	27	4, 443	5, 38
SCA, left nucleus accumbens		Inferior frontal gyrus L	136	−48	45	15	4, 8	6, 01
		Inferior temporal gyrus L	118	−54	−48	−18	3, 73	4, 27

L: left; R: right; PPI: psychophysiological interactions; SCA: seed-based connectivity analysis; MNI—Montreal Neurological Institute.

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
