# Peer review of "Unhealthy yet Avoidable—How Cognitive Bias Modification Alters Behavioral and Brain Responses to Food Cues in Individuals with Obesity"

_nutrients, 2019, doi:10.3390/nu11040874_

Reviewer 1 Report

This is a very interesting and well designed study that examines the affect of CBM training on approach bias to food-images in individuals who are obese. Furthermore, the study reports on the potential neural mechanisms for these effects and addresses a very important theoretical question that is of great interest. 

There are a few minor suggestions below (noting that my expertise lies outside of fMRI research and so I will focus these suggestions on the behavioural elements along with general scientific reporting).

Page 2, Line 52: The authors introduce CBM here and cite relevant research, however I think that the report would benefit from a brief description of what this is at the outset for the non-expert reader.

Page 2, Line 60-61: Here the authors offer methodological differences as as an explanation for discrepancies in previous findings, but it is unclear why and/or how these may affect the findings. A little more detail would be useful here.

Page 2, Lines 79-89: This is a pretty thorough description of the participants, but I think that it is worth including the gender ratio and the age differences between the groups here. I recognise that this information is in the table but this is in the supplementary materials and this sort of information is usually useful to the reader and so should be made more readily accessible.

Page 3, Line 91: Can the authors please clarify if BMI was self-report or objectively measured?

Page 3, Line 98: Typo 'restraing' should read 'restraint' 

Page 3, Line 99: Typo 'to which extend' should read 'to what extent'

Page 3, Line 103: It would be useful to refer to Table S2 here.

Page 4, Lines 125-127: It is unclear at this point why this picture-sorting task is included, a brief explanation would clarify this.

Page 7, Line 226: The authors refer to a definition of regions of interest in the supplementary materials but this is not present. Please include.

Author Response

Response to Reviewer 1 Comments

Point 1: Page 2, Line 52: The authors introduce CBM here and cite relevant research, however I think that the report would benefit from a brief description of what this is at the outset for the non-expert reader.

Response 1: We thank the reviewer for this valuable and constructive comment. As suggested, we have included a brief description of what CBM generally refers to:

Generally, CBM refers to a class of interventions that use experimental paradigms to change biased cognitive processing. This can be done in various domains, such as attention, associations, and behavioral approach or avoidance tendencies."

Point 2: Page 2, Line 60-61: Here the authors offer methodological differences as an explanation for discrepancies in previous findings, but it is unclear why and/or how these may affect the findings. A little more detail would be useful here.

Response 2: Again, we thank the reviewer for this constructive comment, as this information was missing in the manuscript. We have included the following explanation:

“Discrepancies in these results, however, might be explained by differences in samples and stimuli. Approach behaviour towards chocolate or unhealthy food, for example, might not affect normal-weight females the same way as individuals with obesity. In turn, they might be differently affected by a CBM intervention aimed at changing food approach behavior.”

Point 3: Page 2, Lines 79-89: This is a pretty thorough description of the participants, but I think that it is worth including the gender ratio and the age differences between the groups here. I recognise that this information is in the table but this is in the supplementary materials and this sort of information is usually useful to the reader and so should be made more readily accessible.

Response 3: We have now moved Table S1 from the supplementary materials to the manuscripts so that this information is more easily accessible to the reader. We thank the reviewer for pointing this out.

Point 4: Page 3, Line 91: Can the authors please clarify if BMI was self-report or objectively measured?

Response 4: BMI was assessed by measuring height and weight of the participants on the day of the experiment. We thank the reviewer for this comment, as this was not clear in the manuscript. This information is now included in the manuscript.

BMI was calculated before the experiment, after height and weight of the participants were measured by the experimenter”

Point 5: Page 3, Line 98: Typo 'restraing' should read 'restraint' 

Response 5: We thank the reviewer for pointing his out. This is now corrected.

Point 6: Page 3, Line 99: Typo 'to which extend' should read 'to what extent'

Response 6: We thank the reviewer for pointing his out. This is now corrected.

Point 7: Page 3, Line 103: It would be useful to refer to Table S2 here.

Response 7: As suggested, we now refer to the table containing the questionnaire information (Table S1 now, as the table containing participant information is no longer in the supplements).

Point 8: Page 4, Lines 125-127: It is unclear at this point why this picture-sorting task is included, a brief explanation would clarify this.

Response 8: We thank the reviewer for pointing this out, as we realized the motivation for including the picture-sorting was not stated clearly in the manuscript. We have added information in section in 2.2. Behavioral assessment. It now reads as follows:

“Subjects were further asked to sort forty credit card-sized food pictures on a cardboard with a scale (0-10). Subjects were instructed to sort pictures in line with how healthy or unhealthy they perceive them and how much they like or dislike the depicted food items. This measure was included to assess explicit evaluations of food stimuli regarding healthiness and liking. In order to assess whether CBM would affect these ratings, the picture-sorting task was done before and after the experimental paradigm. The picture set was independent of the one used in the fMRI task. It included healthy and unhealthy food pictures of comparable healthiness and liking to the fMRI picture set (see section: Selection of stimuli for details).”

Point 9: Page 7, Line 226: The authors refer to a definition of regions of interest in the supplementary materials but this is not present. Please include.

Response 9: We thank the reviewer for pointing out this mistake. The mentioned section is now placed in the main manuscript file, and the reference to supplementary materials was left by mistake from a previous version of the manuscript. We have cleared the mistake and now refer to section ‘seed definition’ in the main manuscript file.

Reviewer 2 Report

Consider identifying the relevant population in the title. Using "in obesity" to indicate that this work is about obese members of a population makes it unclear whether the paper will be about humans or other animals and has the overall effect of dehumanizing the participants and the population of interest. 

Similarly, it would be helpful to know at a glance that this study is about adults (and hence relevant or irrelevant to various readers). Given the values in Table S1, I assume that it is. 

It appears that there are as many as five hypotheses for this one study. Which is the primary hypothesis?

Table S1 is referred to as "Table S1" throughout the manuscript, but is labeled "Table 1" within the supplementary materials.

I recommend an additional proofreading for occasional issues such as missing punctuation, typographical errors, and a missing instance of "the." 

Line 93: Consider revising "healthiness and liking" the make it clearer that "their" refers to the foods, not the participants. 

99-101: The word "scale" or "scales" is missing. 

Please include more information on the categorization of foods as "healthy" or unhealthy" than, "according to [4]." In particular, I am interested to know whether this refers to foods' perceived healthfulness or unhealthfulness as judged or rated by some participant population or the actual healthfulness or unhealthfulness of the foods. If it is the latter I know that some of my colleagues would be deeply interested to know what expert or experts judged the healthfulness of foods and whether systematic criteria were used. 

Regarding the fMRI task, it would be helpful to have some information about the instructions given to participants for the task. E.g., were participants told to push for horizontal format and pull for vertical format, with the orders subsequently switched? I realize that the task is described in the fourth referenced paper, but it need not be vague here, nor should it be vague how it was modified relative to the fourth referenced paper. 

Author Response

Response to Reviewer 2 Comments

Point 1: Consider identifying the relevant population in the title. Using "in obesity" to indicate that this work is about obese members of a population makes it unclear whether the paper will be about humans or other animals and has the overall effect of dehumanizing the participants and the population of interest. 

Response 1: We thank the reviewer for this valuable comment. We have changed our title accordingly:

Unhealthy Yet Avoidable – How Cognitive Bias Modification Alters Behavioral And Brain Responses To Food Cues In Individuals With Obesity

Point 2: Similarly, it would be helpful to know at a glance that this study is about adults (and hence relevant or irrelevant to various readers). Given the values in Table S1, I assume that it is.

Response 2: We thank the reviewer for this constructive comment. We have made two changes in the manuscript in order to address this and make information on our sample more readily available. In the abstract we have changed “obese people” to “obese adults”. Further, we have moved Table S1 from supplementary materials to the manuscript.

Point 3: It appears that there are as many as five hypotheses for this one study. Which is the primary hypothesis?

Response 3: We thank the reviewer for this comment and question. There was indeed a number of different hypotheses and hypothesised mechanisms that we tested. While it is difficult to state which of these is the primary hypothesis, we tried to differentiate between two levels: the behavioural level (with two subhypotheses), and the neural level (also with two subhypotheses). This differentiation was now added to the manuscript in the introduction section, which now reads:

On the behavioral level, the training was hypothesized to induce two effects: decreased approach tendencies towards unhealthy, and increased approach tendencies towards healthy foods. On the neural level, we hypothesized that CBM training could work through two mechanisms: by a) changing rewarding values of food stimuli and activation in reward-related regions; b) increasing inhibitory abilities and changing activity of brain regions engaged in inhibitory processing and cognitive control. We further explored whether CBM training induces differences in task-independent resting-state functional connectivity.

Point 4: Table S1 is referred to as "Table S1" throughout the manuscript, but is labeled "Table 1" within the supplementary materials.

Response 4: We thank the reviewer for pointing this out. As stated in Response 2, we have moved Table S1 to the manuscript and now refer to it as Table 1. Further, we adjusted the names of all supplementary Tables to fit the main manuscript file.

Point 5: I recommend an additional proofreading for occasional issues such as missing punctuation, typographical errors, and a missing instance of "the."

Response 5: We thank the reviewer for this comment. We scanned the manuscript carefully for such issues and we believe that it now contains no more errors.

Point 6: Line 93: Consider revising "healthiness and liking" the make it clearer that "their" refers to the foods, not the participants.

Response 6: We thank the reviewer for pointing this out. We have revised "healthiness and liking". Further, we have added more information on the picture-sorting task, in line with point 8 of reviewer 1, stating that a more detailed explanation of the picture-sorting task is needed. The section now reads as follows:

“Subjects were further asked to sort forty credit card-sized food pictures on a cardboard with a scale (0-10). Subjects were instructed to sort pictures in line with how healthy or unhealthy they perceive them and how much they like or dislike the depicted food items. This measure was included to assess explicit evaluations of food stimuli regarding healthiness and liking. In order to assess whether CBM would affect these ratings, the picture-sorting task was done before and after the experimental paradigm. The picture set was independent of the one used in the fMRI task. It included healthy and unhealthy food pictures of comparable healthiness and liking to the fMRI picture set (see section: Selection of stimuli for details).”

Point 7: 99-101: The word "scale" or "scales" is missing.

Response 7: We thank the reviewer for pointing out this mistake. We have now added the word ‘scales’ in line 112 (previously 101) after the word ‘system’.

Point 8: Please include more information on the categorization of foods as "healthy" or unhealthy" than, "according to [4]." In particular, I am interested to know whether this refers to foods' perceived healthfulness or unhealthfulness as judged or rated by some participant population or the actual healthfulness or unhealthfulness of the foods. If it is the latter I know that some of my colleagues would be deeply interested to know what expert or experts judged the healthfulness of foods and whether systematic criteria were used.

Response 8: We thank the reviewer for this valuable comment. We realize that including more detailed information regarding picture categorization will add to transparency and comprehensibility of our manuscript. We have therefore revised paragraph 2.5. Selection of stimuli. It now reads as follows:

“Food images for the AAT and the picture-sorting task were selected from the food-pics database [25] and categorized into healthy and unhealthy according to [4]. The categorization was done in line with nutritional values and guidelines, e.g. nutritional tables [27], the Healthy Eating Index [28], or the Dietary Guidelines for American Adherence Index [29]. Healthy food items hereby met at least three of the following five criteria: 1) high nutrient density, 2) low energy density OR high amount of monounsaturated fatty acids, omega-3 fatty acids, or omega-6 fatty acids, 3) high fibre content and/or high water content, 4) low amount of added sugar and 5) low salt content. Unhealthy food items met at least one of the following criteria: 1) low nutrient density combined with high energy density, 2) High energy density AND an unfavorable ratio of saturated fatty acids, omega-3 fatty acids, or omega-6 fatty acids and 3) high amount of added sugar. Further, a pre-rating online survey was conducted in order to validate nutritional categorization from a subjective perspective. Participants (n=98, BMI range from 18 to 36 kg/m2) were asked to rate food pictures on a scale from 0 to 10 (0=unhealthy, 10=healthy) regarding how healthy or unhealthy they perceived them to be. Only images that were clearly identified as healthy or unhealthy were included (mean ratings between 0 and 2 or 8–10). This way, pictures were categorized as being healthy or unhealthy according to both, objective nutritional and subjective criteria.”

Point 9: Regarding the fMRI task, it would be helpful to have some information about the instructions given to participants for the task. E.g., were participants told to push for horizontal format and pull for vertical format, with the orders subsequently switched? I realize that the task is described in the fourth referenced paper, but it need not be vague here, nor should it be vague how it was modified relative to the fourth referenced paper

Response 9: We thank the reviewer for this comment; we agree that adding more information about the task also adds to the clarity of the manuscript. We have edited the ‘fMRI task’ section of our manuscript, which now reads as follows:

Participants react with push and pull movements of a joystick to the format of presented food pictures (push-vertical/pull-horizontal, or push-horizontal/push-vertical). The association between the picture format and joystick movement was randomly assigned to each participant and did not change throughout the task. The AAT consisted of three main phases: a pre-phase, a training or a sham-training phase, and a post-phase. In the pre-, post- and sham-training phases food pictures were presented equally often in push and pull formats. In the training phase 90% of unhealthy pictures appeared in a push format, and 90% of healthy pictures appeared in a pull format (Figure 1B). In this respect there was no modification of the task as compared to [4].’